# *Penguin*: Parallel-Packed Homomorphic Encryption for Fast Graph Convolutional Network Inference

**Ran Ran**
North Carolina State University
`rran@ncsu.edu`

**Nuo Xu**
Lehigh University
`nux219@lehigh.edu`

**Tao Liu**
Lawrence Technological University
`tliu3@ltu.edu`

**Wei Wang**
Anonym, Inc.
`wei@anonymco.com`

**Gang Quan**
Florida International University
`gaquan@fiu.edu`

**Wujie Wen**
North Carolina State University
`wwen2@ncsu.edu`

## Abstract

The marriage of Graph Convolutional Network (GCN) and Homomorphic Encryption (HE) enables the inference of graph data on the cloud with significantly enhanced client data privacy. However, the tremendous computation and memory overhead associated with HE operations challenges the practicality of HE-based GCN inference. GCN inference involves a sequence of expensive matrix-matrix multiplications, and we observe that directly applying the state-of-the-art HE-based secure matrix-matrix multiplication solutions to accelerate HE-GCN inference is far less efficient as it does not exploit the unique aggregation mechanism of two-dimension graph node-features in GCN layer computation. As a result, in this paper, we propose a novel HE-based ciphertext packing technique, i.e.,*Penguin*, that can take advantage of the unique computation pattern during the HE-GCN inference to significantly reduce the computation and memory overhead associated with HE operations. Specifically, *Penguin* employs (*i*) an effective two-dimension parallel packing technique for feature ciphertext with optimal graph node partitioning and graph feature interleaving, and (*ii*) an interleaved assembly technique that can effectively make use of blank slots to merge ciphertexts after feature reduction and thus significantly reduce costly rotation operations. We perform detailed theoretical analysis to support our arguments. In the meantime, our experimental results also show that *Penguin* can achieve up to $\sim 10\times$ speedup and around $\sim 79\%$ reduction in computational memory overhead, significantly outperforming state-of-the-art solutions. To the best of our knowledge, this is the first work that can ensure the protection of both graph structure and features when accelerating HE-GCN inference on encrypted data. Our code is publicly available at `https://github.com/ranran0523/Penguin`.

## 1 Introduction

Graph Convolution Neural Networks (GCNs) have recently demonstrated phenomenal performance for many privacy-sensitive applications such as social networks [34], cross-domain recommendation systems [35], and personal healthcare [18]. A popular solution for clients seeking to leverage these advanced GCN models is to utilize cloud-based inference services. However, clients often hesitate to share their graph data with the public cloud due to concerns about sensitive information, such as graph structure and node features that reveal personal social relationships and medical records. To address this privacy concern, one viable approach is to adopt the Homomorphic Encryption (HE) scheme [3, 6, 7]. By performing the entire inference computation on the cloud using encrypted data,

37th Conference on Neural Information Processing Systems (NeurIPS 2023).

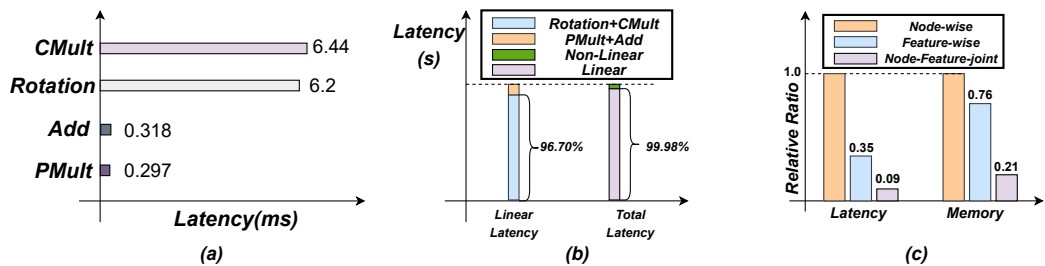

Figure 1: (a) Latency comparison of different HE operations under same encryption parameter and hardware environment; (b) Latency breakdown of linear/nonlinear HE operations in a typical GCN layer computation. (Detailed settings in Sec. 4.1). (c) Single optimization and wasted ciphertext slots have a negative effect on memory utilization and computation latency.

the privacy of client data is significantly enhanced. This enables privacy-preserving GCN inferences while ensuring that sensitive information remains confidential.

While the idea of embedding HE into GCN inference on graph data seems appealing, it faces several significant challenges: **Firstly,** similar to HE-based CNN inference on non-graph data (such as for using convolutional neural networks (CNN) [10, 4, 8, 23, 2, 19, 26, 17, 14]), the enhanced privacy would come at the cost of the tremendously escalated computational overhead associated with HE operations (e.g., ciphertext (ct) rotations/multiplications, additions), which could be orders of magnitude higher than the counterparts in the non-encrypted computation [10, 29, 16]. **Secondly,** existing solutions focusing on alleviating computation overhead of HE-based CNN inference may not be applicable or optimal to GCNs due to computing pattern differences between the CNN and GCN [21]. For example, a GCN layer's computation is dominated by the special consecutive matrix multiplications ($A \cdot X \cdot W$) for 2-dimensional feature-node aggregation–feature aggregation via multiplying a high dimensional feature matrix $X$ with weight matrix $W$, followed by graph node aggregation with $A$, while a CNN layer's computation is bottlenecked by multi-channel 2D convolutions. **Thirdly,** simply treating the above critical matrix operations in HE-based GCN inference as a traditional encrypted matrix-matrix multiplication (MM) problem for speedup is sub-optimal because: 1) state-of-the-art (SOTA) HE-based MM acceleration often requires the matrix to satisfy some special properties, e.g. square matrix with size $64 \times 64$ [15], while GCN matrices like feature matrix $X$ are typically irregular depending on applications (i.e. $2708 \times 1433$ in Cora dataset [32]); 2) SOTA solutions focus on a one-time MM without considering the consecutive MMs incurred by the two-dimensional feature-node aggregation, as well as the further processing of MMs' result in the next GCN layer. This leads to inefficient ciphertext space utilization and unnecessary HE operations, which further translates into prolonged HE-GCN inference, as we shall show in Sec. 4.2.

To better understand the computation cost of HE operations that dominate the HE-GCN inference latency, we profile the latency of different HE operations using one GCN layer with 32 hidden units and the Cora dataset with 2708 graph nodes and 1433 (32) input (output) features per node. All HE operations are defined in Sec. 2. For generality, we assume *both feature matrix and adjacency matrix are encrypted*, which is a typical case in inductive learning (e.g. dynamic graph structure in link prediction) [22]. Without loss of generality, the same indexed features from different nodes are packed as a ciphertext (feature-wise packing) and the encrypted matrices are diagonal-encoded for MMs (detailed settings in Sec. 4.1). As Figure 1 (a) shows, first, the latency of ciphertext rotation and ciphertext multiplication (CMult) can be much higher than other operations like plaintext (pt) multiplication (PMult) or Addition, e.g. $> 20\times$ Rotation v.s. PMult. Furthermore, about $> 99\%$ latency comes from the linear operations (mainly HE rotation and CMult due to the consecutive MMs), instead of the nonlinear operations (ReLU replaced by a square function) due to feature reduction in GCN (from 1433 input features to 32 output features). Meanwhile, for linear latency, Rotation and CMult dominate the latency (e.g. $> 96\%$ of total) as the size of the adjacency matrix could be quite large (Cora: $2708 \times 2708$) in the GCN problem. Last, we profile the latencies of different ciphertext packing formats under the same evaluation setup as (b) in Figure 1 (c). From the profiling result in Figure 1 (c), either the node-wise packing format (e.g. 1 ciphertext contains one node's 1433 features) or the feature-wise packing format (e.g. 1 ciphertext contains the same indexed features from 2707 nodes) could not effectively perform the HE-GCN inference. With node-feature-joint

packing format (e.g. 1 ciphertext packs 32 features and 128 nodes) by our proposed Two-Dimension Parallel-Packing (see Sec. 3.2), the ciphertext size is fully exploited, and the total HE operation count reaches a minimum, leading to significantly reduced latency and memory cost. These results indicate that the key to accelerating the HE-based GCN inference is to significantly reduce the rotation and CMult operations with a GCN-dedicated ciphertext packing format.

To this end, we propose *Penguin*, a novel HE ciphertext packing framework dedicated to accelerating GCN inference with the consideration of **encrypting both graph structure and features simultaneously** (both adjacency matrix $A$ and input feature matrix $X$). The driving vision of *Penguin* is: *feature ciphertext packing ($X$) for efficient HE-based GCN inference needs to be designed in a manner that is aware of the unique GCN computation–both the left-side graph node aggregation $AX$ and right-side feature aggregation ($XW$), instead of optimization in one direction (either $AX$ or $XW$). In this way, the whole ciphertext space can be efficiently utilized with minimized slot waste, enabling the significant reduction of ciphertext number (memory overhead) as well as the expensive HE rotation and CMult operations under the single instruction multiple data (SIMD) architecture.* Our major contributions are three-fold: **1)** We propose an efficient two-dimension parallel packing technique for ciphertext via optimal graph node partition and feature interleaving. By performing the feature-level aggregation first and formulating the HE computation overhead as a constrained optimization problem, we analytically obtain the best feature-node partition that can maximize the usage of ciphertext space and minimize the costly HE operations. Experimental results are well consistent with theoretical analysis. **2)** We propose an interleaved assembling (IA) technique to efficiently merge ciphertexts with blank slots incurred by feature dimension reduction in the feature aggregation stage. This extra-level optimization further significantly reduces the number of ciphertexts and associated HE operations. **3)** We comprehensively evaluate our proposed *Penguin* for CKKS-based GCN inference using Cora-based graph node classification, Citeseer-based link prediction, and Pubmed-based link prediction. Results show that our method achieves by up to about $10\times$ inference speedup and $79\%$ memory overhead reduction, significantly outperforming the state-of-the-art solutions. *To the best of our knowledge, this is the first work focusing on accelerating the HE-based private graph convolutional neural network inference on encrypted graph data, of which both the sensitive graph features and graph structure are protected.*

## 2   Preliminary

**CKKS Homomorphic Encryption Scheme.** Homomorphic Encryption (HE) allows computations on encrypted data. HE has different categories according to the different computation types they support. The Leveled HE (LHE) schemes support a limited number of additions or multiplications while Fully HE (FHE) allows an arbitrary number of computations using a bootstrapping procedure that can effectively refresh the ciphertext and obtain a new ciphertext that encrypts the same value but has lower noise [9]. In this work, we focus on reducing the number of bottlenecked operations in CKKS–one of the promising LHEs, without considering the costly bootstrapping.

CKKS [6] is an LHE scheme and its security is based on the hardness of ring learning with errors (RLWE) problem. CKKS allows arithmetic operations on encrypted data over fixed-point numbers with predefined precision, which makes it an ideal candidate for performing machine learning tasks where most of the computations are approximate. The supported homomorphic operations include ciphertext addition *Add* $\sim (ct_1 + ct_2)$, ciphertext multiplication *CMult* $\sim (ct_1 \times ct_2)$, plaintext multiplication *PMult* $\sim (ct \times pt)$, ciphertext *Rotation* $\sim \rho(ct, k)$. The rotation is to apply Galois automorphisms of the cyclotomic extension to the plaintext polynomials in encrypted form resulting in a cyclic shift of the slot vector. Among these four operations, *Rotation* and *CMult* are substantially slower ($\sim 20\times$ slower) than ciphertext-plaintext addition and multiplication as shown in our runtime performance of CKKS in Figure 1 due to the expensive key-switching operation [23].

**Graph Convolution Neural Network.** To extract the hidden graph features $H$, the 2-dimensional feature-node aggregation of a typical GCN layer can be often abstracted as [21]:

$$H = \sigma(\tilde{D}_j^{-\frac{1}{2}} \tilde{A}_j \tilde{D}_j^{-\frac{1}{2}} XW) \tag{1}$$

Where $X \in R^{N \times F}$ is the input feature matrix. $W_j \in R^{F \times F'}$ represents weight parameters to transform the input features from an input dimension $F$ to an output dimension $F'$ ( feature level aggregation). $\tilde{D}_j$. $\tilde{A}_j$ is the adjacency matrix with self-loop. The $XW$ term is implemented by

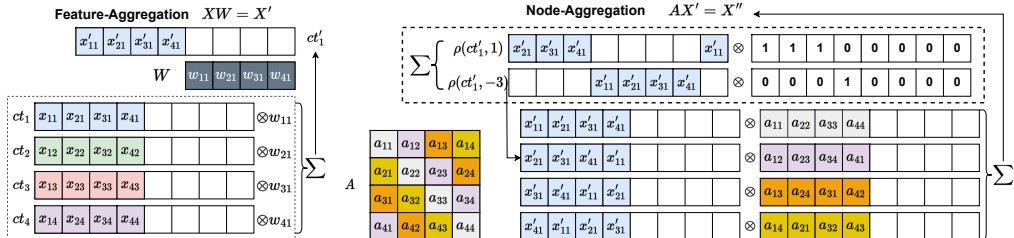

Figure 2: Feature-Optimized Packing Ciphertext Computation Flow.

a fully-connected (FC) layer (node level aggregation) and then multiplied with the normalized adjacency matrix $\tilde{D}_j^{-\frac{1}{2}} \tilde{A}_j \tilde{D}_j^{-\frac{1}{2}}$. Finally, a non-linear activation function $\sigma$ (e.g. ReLU) is applied to get one GCN layer's output feature matrix $H$. Throughout this work, we refer $A$ as the normalized adjacency matrix since normalization could be absorbed in a pre-processing step.

**Threat Model.** We adopt a threat model setting consistent with prior works [10, 15, 4, 8, 19, 23, 28]. A client uploads private and sensitive data to the cloud for obtaining the online machine learning model prediction results. The cloud server is semi-honest (e.g. honest but curious). To ensure data privacy, the client encrypts their own data by HE and decrypts this inference result by their private key. In this work, we focus on encrypting both graph node features $X$ and the normalized adjacency matrix $A$. The clients run the decoder of GAE [22] at their end because this step does not involve trained model parameters on the cloud server.

## 3 Method

**Overview.** The GCN inference $A \cdot X \cdot W$ can be separated into the two-dimension ($A \cdot X$ on the nodes and $X \cdot W$ on the features) aggregation on feature matrix $X$. When we perform HE matrix multiplication on the encrypted feature matrix (ciphertexts), it is inevitable that we need to perform HE rotation on the same ciphertext. Unfortunately, the rotation operation not only incurs high latency but also generates a huge number of ciphertext copies that consume a large amount of memory space. In this section, we propose a holistic solution set to systematically address these issues.

In order to effectively reduce the number of ciphertexts involved in HE computation, our design is built upon the feature-wise packing since multiplying $W$ often leads to a lower feature dimension. However, for non-densely packed ciphertexts, feature-wise packing is further subject to the data alignment issue, resulting in extra rotations. To overcome this challenge, we propose the two-dimension parallel-packing. In addition, considering that the layer-wise feature number reduction would result in many wasted slots, we further propose the interleaved assembling to efficiently merge such ciphertexts.

### 3.1 Motivation of Feature-Oriented Ciphertext Packing

The major inference computation in GCN can be illustrated as $A \cdot X \cdot W$, where $A \in R^{N \times N}$ is the normalized adjacency matrix used for node-wise aggregation, $X \in R^{N \times F}$ is the input feature matrix, and $W \in R^{F \times F'}$ is the weight matrix used for feature-wise aggregation. Apparently, we can choose $A \cdot X$ or $X \cdot W$ as the first step, which will not change the final product. However, considering that matrix $X$ is encrypted as ciphertexts, the order of computation will affect the efficiency since the ciphertexts with fewer dimensions will reduce the required HE operations and copies of the ciphertexts. For example, if $F' < F$, we first perform $X \cdot W$ to produce an intermediate product with fewer dimensions, i.e., $R^{N \times F'}$. This will reduce the computational overhead and latency in the next step $A \cdot X$. On the contrary, if $F < F'$, we first perform $A \cdot X$.

We explore two ciphertext packing design options that could lead to minimized computational overhead of a single-dimension aggregation (either graph node or feature). One is the feature-wise packing, where one ciphertext only packs one feature data from different nodes. The number of ciphertexts is proportional to feature number. The other is the node-wise packing, where the number of ciphertexts is equal to the number of graph nodes. However, in this case, the number of graph

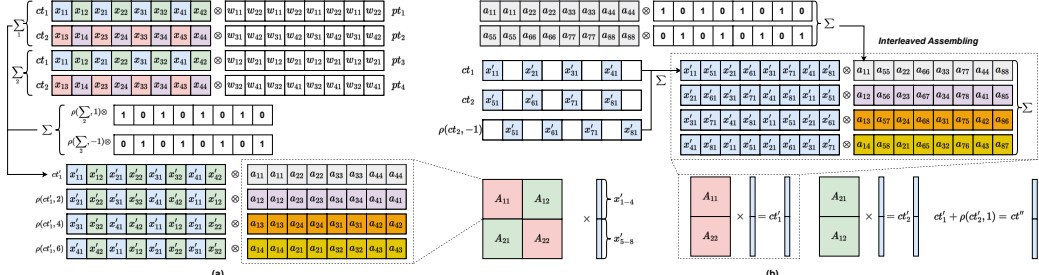

Figure 3: (a) The Two-Dimension Parallel-Packing. (b) The Interleaved Assembling.

nodes (or ciphertexts) does not change during inference, this inevitably results in too many wasted empty slots in the ciphertexts and thus would yield more HE rotations. As the example in Figure 2 shows, we assume the ciphertext packing size is 8, adjacency matrix $A \in R^{4 \times 4}$, weight matrix $W \in R^{4 \times 1}$, and feature matrix $X \in R^{4 \times 4}$. The 4 feature-wise packing ciphertexts can be reduced to 1 ciphertext after feature aggregation, which only needs 4 ciphertext-multiplication (CMult) in the next adjacency matrix multiplication. If using the node-wise packing, we still need 16 CMult in the next step. Therefore, we choose feature-wise packing in our design.

### 3.2 Two-Dimension Parallel-Packing

**Optimization Problem Formulation**  Following the definition in the previous section, we assume that the ciphertext has $M$ available slots for packing data and consider the following general case: $M > N$ and $M > F$. As shown in Figure 2, after the feature-wise packing, the same features of different data from feature matrix $X$ are encoded into the same ciphertext $X = [ct_1 \ldots ct_F]$ and then multiplied with weights $w_{ij}$ to get matrix $X' = XW = [ct'_1, \ldots, ct'_{F'}]$ in the same feature-wise packing format (see Eq.( 2)).

$$ct'_j = \sum_{i=1}^{F} ct_i \otimes w_{ij}, j \in F' \tag{2}$$

For node-wise aggregation, we need to perform diagonal-encoded matrix multiplication [12] on ciphertexts $F'$ individually. However, due to $M > N$, we need to generate the corresponding ciphertext copies with each data aligned by rotating each ciphertext twice and scaling with the mask vector then summing (see Eq.( 3)).

$$ct'_{ij} = \rho(ct'_j, i) \otimes ms_i + \rho(ct'_j, -(N-i)) \otimes ms_{-(N-i)}, i \geq 1 \tag{3}$$

After that, we multiply them with the corresponding diagonal-encoded vector $d_i$ of adjacency matrix $A$ and sum up them to get the node-wise aggregation result of $A \cdot X'$ (see Eq.( 4)).

$$ct''_j = \sum_{i=0}^{N-1} (ct'_{ij}) \otimes d_i, ct'_{0j} = ct'_j \tag{4}$$

In this process, we can find that data alignment issue for non-densely packed ciphertexts leads to extra rotations. We propose the two-dimension parallel-packing to solve it. Our idea is to leverage the matrix partition to fully pack data in all slots and amortize the HE computation cost.

Figure 3(a) shows our basic idea. We partition $n$ (a power two number) graph nodes into a small block to fully utilize the size of the ciphertext and encode the feature matrix $X$. In each ciphertext, we actually pack $n$ nodes corresponding to $f$ different features together in an interleaved way. For feature-wise aggregation, we adopt the baby-step algorithm [19, 15] to get the different output features with good alignment. As shown in Figure 3 (a), the number of rotations $2(f-1)$ used here for each ciphertext depends on the number of different features $f = M/n$ (we assume the output feature $F' \geq f$). Then, we continue to rotate each ciphertext for $n - 1$ times and perform the diagonal encoded matrix multiplication [12] for $A \cdot X$. The total complexity of rotation is:

$$(n-1) \cdot (N \cdot F'/M) + (2(f-1)) \cdot (N/n \cdot (F'/f)) \tag{5}$$

Since $M = n \cdot f$, the total complexity is further equal to

$$(N \cdot F'/M) \cdot ((n-1) + 2(f-1)) \tag{6}$$

According to Cauchy-Schwarz inequality:

$$O(n + 2f) \geq O(2 \cdot \sqrt{2nf/2}) = O(2 \cdot \sqrt{M}) \tag{7}$$

Where $n = 2f$, the total complexity of rotation reaches a minimum. Hence minimizing the number of rotations can be modeled as an optimization problem:

$$argmin_{(f,n)}\{(n-1) + 2(f-1)\} s.t. \begin{cases} M = n * f \\ f = 2^k, k \in N^+ \\ M > N, M > F \end{cases} \tag{8}$$

**Supporting Large Graph**  Eq. 8 assumes that the number of graph nodes should be smaller than that of ciphertext slots ($M > N$), however, for scaled graph networks, it is possible that $M \leq N$. For example, the PubMed [32] contains 19717 nodes, which is far more than the 4096 ciphertext slots. To address this, our method can be scalable to such cases by splitting a large graph into several sub-graphs. Assuming we use feature-wise encoding and it requires 5 cts (each with 4096 slots) to pack 1 feature. Accordingly, each feature will have 4 cts with fully packed 4096 nodes and 1 ct with partially packed 3333 nodes. To solve the problem under the constraint–$M \leq N$, we split $N$ as:

$$N = x \cdot M + R \tag{9}$$

where $x = N \bmod M$, $R = N \% M$. Eq. 9 leads to one $R \times F$ sub-block matrix and $x$ of $M \times F$ sub-block matrices. For the $R \times F$ matrix, we refer to Eq. 7 to optimize the $Rotation$ as $N = R < M$. For other $M \times F$ sub-block matrices with $N = M$, we change the assumption from $N < M$ to $N = M$. Then, again with our proposed Two-Dimension Parallel-Packed ct, the total complexity of rotations becomes:

$$(M \cdot F'/M) \cdot ((n-1) + 2(f-1)) = F' \cdot (n + 2f - 3) \tag{10}$$

The corner case $f = 1$ is different from that in the discussion of optimization problem formulation. Because ct here is fully packed for $n = M$ and does not have the data alignment issue. For $n = M, F = 1$, the total complexity of $Rotation$ becomes:

$$F' \cdot (M - 1) \tag{11}$$

Except for this corner case $n = M, f = 1$, the total complexity reaches a minimum when $n = 2f$. We compare the previously proved minimum with the corner case $n = M, f = 1$ here, and get the difference of rotations complexity as follows:

$$F' \cdot ((M-1) - (2\sqrt{M} - 3)) = F' \cdot (M - 2\sqrt{M} - 4) = F' \cdot (\sqrt{M} - 2)^2 > 0 \tag{12}$$

In general, since $M$ is set as $\geq 2^{11}$ to guarantee security level [17, 28], the above inequality 12 always holds. Thus, when considering a $M \times F$ matrix, the proposed Two-Dimension Parallel-Packing can still reach the minimum at $n = 2f$.

### 3.3   Interleaved Assembling

In GCN inference, the reduction in feature size may result in wasted slots in two-dimension parallel-packing. As the example shown in Figure 3(b), we optimally encode 32 different features into one ciphertext at the beginning. After the feature extraction layer with 16 hidden units, the previous dense encoded ciphertext will have half of the slots turn to blank. These blank slots in ciphertext bring higher memory overhead, especially given that the adjacency matrix $A$ is also encrypted, resulting in more CMult operations thus computational overhead. We propose interleaved assembling to solve this issue. Figure 3(b) shows our idea. We rotate the $ct_2$ that contains node 5-8's features by 1 slot and then add it with $ct_1$ that contains node 1-4's features. After that, we have a new ciphertext $ct'$ that contains 8 nodes with 1 feature. Meanwhile, we multiply the two mask vectors with the ciphertexts for sub-square matrix $A_{11}$ (for node 1-4) and $A_{22}$ (for node 5-8) and get an interleaved assembled ciphertext contains matrix $A_{11}$ and $A_{22}$. Then, by rotating ciphertext $ct'$ 3 times and performing element-wise multiplication with new diagonal-encoded ciphertexts of the matrix $A_{11}$ and $A_{22}$, we could get the results–ciphertext $ct'_1$ of node 1-4 with the matrix $A_{11}$ and node 5-8 with the matrix $A_{22}$ simultaneously. In this way, the complexity of HE operations including both rotation and ciphertext multiplication can be reduced by half. After that, we repeat the steps to perform multiplication on $ct'$ and ciphertext that contains $A_{21}$ and $A_{12}$ to get the $ct'_2$. Based on the formula $ct'' = ct'_1 + \rho(ct'_2, 1)$, we get the final result ciphertext $ct''$ that multiples matrix $(A_{11}, A_{12}, A_{21}, A_{22})$. By leveraging such an interleaved assembling, we could achieve $\frac{f'}{f}$ times reduction of the total computational complexity, where $f'$ is the number of features on the current ciphertext, and $f$ is the number of features on the ciphertext before feature reduction.

# 4 Evaluation

## 4.1 Experiment Setup

**Datasets.** We adopt the Cora [32], Citeseer [11] and Pubmed [32] scientific publication datasets for graph learning. The Cora, Citeseer, and Pubmed contain 2708, 3327, and 19717 publication nodes divided into 7, 6, and 3 classes respectively. And each node consists of 1433, 3703, and 500 unique word features, respectively. To test the link prediction task [22], 90% of edges are removed and all node features are retained on all datasets.

**Models.** We train 3 Graph Auto-Encoder (GAE) models with 2 hidden layers and 2 activation layers on 3 different datasets, i.e., Cora, Citeseer, and Pubmed. The three models follow the same GAE architecture in [22], and are implemented using the DGL library [33]. Table 1 lists the model architecture and pertinent encryption parameters for encrypting both adjacency matrix $A$ and feature matrix $X$. We use $x^2$ as the non-linear function [10] to replace the ReLU activation and apply the ADAM optimizer to train the model for 200 epochs using a learning rate of 0.01. The accuracy of each model (AUC in Link Prediction) is maintained at the original level.

**Encryption parameters.** For all tasks, we apply a scaling factor $\Delta = 2^{30}$ to ensure the accuracy of the encrypted inference using CKKS. Each rescale consumes 30 bits of ciphertext modulus $Q$, and there are 6 times rescale and corresponding 6 levels across the whole network. Thus, we set $Q = 218$, and the polynomial degree $N = 2^{13}$ to guarantee a 128-bit security level. Additionally, the scale factor of mask plaintext used in comparison with E2DM [15] & uSCORE [13] is set to $2^{15}$.

**Baseline designs.** To better evaluate the proposed approach, we develop several baselines, including:

- Penguin-family. We implement several Penguin baselines by applying only our proposed two-dimension parallel-packing technique (see Sec. 3.2). We set up different pairs of features and nodes when optimizing the packing format. Table 2 lists the numbers of features/nodes selected. Here $Penguin(f, n)$ denotes that $f$ features and $n$ nodes are used in the corresponding baseline design. Note that, the baseline designs with $f = 1$ or $n = 1$ are the extreme cases when only the feature-wise or node-wise packing method is used.

- Penguin+IA. We develop two Penguin+IA baselines by further applying the proposed Interleaved Assembling (IA) technique (see Sec. 3.3) to the Penguin-family.

- We also implement a set of state-of-the-art secure matrix matrix multiplication solutions, which include Gazelle [17], HElayers [1], E2DM [15] and uSCORE [13]. To ensure a fair comparison, we do not apply the activation pruning technique for multiplicative level reduction across all implementations [28], as this technique is orthogonal to our proposed solutions.

**Measurements.** We use inference latency as our main performance metric, which is averaged over 20 simulations. Besides, we record the Homomorphic Operation Count (HOC), including the number of rotations (*Rotation*), the number of ciphertext multiplications (*CMult*), etc. We also calibrate the numbers of ciphertexts and memory usage. A lower number of these metrics indicates better performance.

**Environment.** We conduct all experiments on a machine equipped with Threadripper 3975WX CPU using the single thread setting to test the inference latency and train these GAE models with 2 Nvidia 3090 GPUs. We use Microsoft SEAL version 3.7.2 [31] to implement the RNS-variant of CKKS [5] scheme.

Table 1: Model and encryption parameters.

| Dataset | # Layers | | | Accuracy | Encryption Parameters | | | Mult | Security |
|---------|----------|----------|------------|----------|---|---|---|-------|----------|
| | Hidden1 | Hidden2 | Activation | (AUC) | N | Q | P | Level | Level |
| Cora | | | | 0.974 | | | | | |
| Citeseer | 32 | 16 | $x^2$ | 0.747 | 8192 | 218 | 30 | 6 | 128-bit |
| PubMed | | | | 0.858 | | | | | |

Table 2: Ablation study of Two-Dimension Parallel-Packing and Interleaved Assembling.

| Dataset | Packing-Format | HOC | | Others | # of Ciphertexts | Memory (GB) | Latency (s) | Speedup (×) |
|---|---|---|---|---|---|---|---|---|
| | | Rot | CMult | | | | | |
| Cora | Penguin(1433,1) | 1048K | 74K | 282K | 2708-2708-2708 | 2.38 | 7018.51 | - |
| | Penguin(1, 2708) | 260K | 130K | 223K | 1433-32-16 | 1.82 | 2475.78 | 2.83 |
| | Penguin(16,256) | 9.7K | 9.3K | 157k | 990-22-11 | 0.49 | 678.03 | 10.35 |
| | **Penguin(32,128)** | **8.3K** | 124K | 188K | 990-22-22 | 0.65 | 871.15 | 8.06 |
| | Penguin(64,64) | 13.5k | 237k | 365K | 990-43-43 | 1.25 | 1650.28 | 4.25 |
| | **Penguin(32,128)+IA** | **6.9K** | 9.3K | 157K | 990-22-11 | 0.49 | **660.67** | **10.62** |
| | Penguin(64,64)+IA | 10.3K | 9.3K | 220k | 989-22-11 | 0.49 | 693.13 | 10.13 |
| Citeseer | Penguin(3703,1) | 1521K | 1110K | 3852K | 3327-3327-3327 | 2.92 | 9240.10 | - |
| | Penguin (1, 3327) | 319K | 160K | 385K | 3703-32-16 | 3.08 | 3064.91 | 3.01 |
| | Penguin(16,256) | 110K | 130K | 324K | 3016-26-13 | 1.40 | 950.30 | 9.72 |
| | **Penguin(32,128)** | **9.8K** | 173K | 367K | 3016-26-26 | 1.62 | 1225.05 | 7.54 |
| | Penguin(64,64) | 16.3K | 346K | 734K | 3016-52-52 | 2.51 | 2429.47 | 3.80 |
| | **Penguin(32,128)+IA** | **7.4K** | 130K | 324K | 3016-26-13 | 1.39 | **928.10** | **9.96** |
| | Penguin(64,64)+IA | 12K | 130K | 387K | 3016-26-13 | 1.39 | 982.08 | 9.41 |
| PubMed | Penguin(19717,1) | 5974K | 817K | 12687K | 19717-19717-19717 | 9.75 | 44586.03 | - |
| | Penguin (1, 500) | 1106K | 4732K | 4897K | 2500-160-80 | 17.3 | 37727.58 | 1.18 |
| | Penguin(16,256) | 69K | 4673K | 4837K | 2496-156-78 | 3.76 | 30906.28663 | 1.44 |
| | **Penguin(32,128)** | **59K** | 6151K | 6314K | 2480-155-155 | 11.6 | 40474.11 | 1.10 |
| | Penguin(64,64) | 117K | 12222K | 12547K | 2472-309-309 | 42.9 | 80424.70 | 0.55 |
| | **Penguin(32,128)+IA** | **49K** | 4633K | 4794K | 2480-155-78 | 3.76 | **30522.43** | **1.46** |
| | Penguin(64,64)+IA | 73K | 4633K | 4954K | 2472-155-78 | 3.75 | 30701.59 | 1.45 |

## 4.2 Evaluation Results

### 4.2.1 Two-Dimension Parallel-Packing

Table 2 presents our evaluation results of the proposed two-dimension parallel-packing and interleaved assembling approach. We find that the packing format $Penguin(f, n = 1)$ performs the worst on the three datasets due to having the largest number of HOCs and no slot packing optimization. This results in significant latency and memory overhead. In particular, since PubMed contains more encrypted features (number of cts), the same design performs worse on PubMed than on the other two datasets. We use this $Penguin(f, n = 1)$ as the baseline to compare the speed of other approaches.

Our results clearly show that our proposed two-dimension parallel-packing method can significantly reduce the HOCs (especially the number of rotations) and the number of ciphertexts. For example, on the Cora dataset, our $Penguin(16, 256)$, $Penguin(32, 128)$, and $Penguin(64, 64)$ designs can reduce the number of rotations from 1048K to 9.7K, 8.3K, and 13.5K, respectively, thus reducing memory usage by $\sim 79\%$, $\sim 76\%$, and $\sim 47\%$, and reaching $\sim 10.35\times$, $\sim 8.06\times$, and $\sim 4.25\times$ speed up, respectively.

In particular, the results we observed are well consistent with the theoretical analysis. For example, with $M = f * n = 4096, n = 2f$, we have the theoretical minimum $f_{min} = \sqrt{2048} \simeq 45$ (see Section 3.2). We can observe that the baseline $Penguin(32, 128)$ with $f = 32, n = 128$ is very close to the theoretical minimum and achieves the best results among the three designs. Meanwhile, other HOCs besides Rotation may increase under the optimal packing and affect the overall latency. For example, $Penguin(32, 128)$ yields more ciphertext multiplication (CMult) than $Penguin(16, 256)$ due to wasted slots from feature reduction, which can be further optimized using the proposed Interleaved Assembling method.

Moreover, as our discussion of large graph in Section 3.2 indicates, all designs exhibit significantly worse performance in the PubMed dataset compared to the other two datasets. This is because the number of nodes in PubMed is significantly larger than the size of ciphertext ($19717 \gg 4096$), which means that it needs to be multiplied with a large $19717 \times 19717$ adjacency matrix. Therefore, the number of CMult $\gg$ number of Rot. However, our proposed packing technique can still improve the performance in such cases.

### 4.2.2 Interleaved Assembling

Table 2 also reports the evaluation results of incorporating the two-dimensional parallel packing and interleaved assembly methods. For example, in Cora, the number of rotations, the number of CMult, and the number of other HOCs in the $Penguin(32, 128) + IA$ design are further reduced by 1.4K, 114.7K, and 31K, respectively, compared to the parallel-packing only $Penguin(32, 128)$. This makes $Penguin(32, 128) + IA$ the best design on all datasets, i.e., with the minimum memory

Table 3: Compare with the state-of-the-art.

| Dataset | Method | Security Level | Latency (s) | Amortized Latency | Speedup ($\times$) |
|---------|--------|----------------|-------------|-------------------|---------|
| Cora | Gazelle [17] | 128-bit | 3832.36 | 1.42 | - |
| | E2DM(64) [15] | 98-bit | 3150.74 | 1.16 | 1.22 |
| | HElayers [1] | 128-bit | 2102.47 | 0.78 | 1.82 |
| | uSCORE(32,128) [13] | 98-bit | 1727.12 | 0.64 | 2.22 |
| | **Penguin(32,128)+IA** | 128-bit | **660.57** | **0.24** | **5.92** |
| Citeseer | Gazelle [17] | 128-bit | 4727.94 | 1.42 | - |
| | E2DM(64) [15] | 98-bit | 4561.15 | 1.37 | 1.04 |
| | HElayers [1] | 128-bit | 3044.58 | 0.92 | 1.54 |
| | uSCORE(32,128) [13] | 98-bit | 2377.50 | 0.72 | 1.97 |
| | **Penguin(32,128)+IA** | 128-bit | **928.10** | **0.28** | **5.07** |
| Pubmed | Gazelle [17] | 128-bit | 158655.54 | 8.05 | - |
| | E2DM(64) [15] | 98-bit | 154530.49 | 7.84 | 1.03 |
| | HElayers [1] | 128-bit | 103283.56 | 5.24 | 1.54 |
| | uSCORE(32,128) [13] | 98-bit | 78843.49 | 4.00 | 2.01 |
| | **Penguin(32,128)+IA** | 128-bit | **30522.43** | **1.55** | **5.19** |

usage of 0.49GB, 1.39GB, and 3.76GB and a $10.62\times$, $9.96\times$, and $1.46\times$ speedup on dataset Cora, Citeseer, and PubMed, respectively. These results illustrate that our proposed interleaved assembly can effectively reduce the wasted empty slots and save the number of ciphertexts selected in the computation, thus significantly improving the efficiency based on the SIMD.

### 4.2.3 Compare with State-of-the-art Solutions

In our evaluation, we conduct a comparative analysis of our best-designed model, $Penguin(32, 128) + IA$ with several state-of-the-art (SOTA) solutions, including Gazelle [17], HElayers [1], E2DM [15] and uSCORE [13]. All these SOTA solutions can speed up HE-GCN inference with the optimized matrix-matrix multiplication. The results are summarized in Table 3.

It is worth noting that our approach uses encryption parameters which offer a 128-bit security level. This security level is higher than that of the other two SOTA solutions, which can only guarantee a 98-bit security level and require more multiplicative levels to mask the plaintexts. For a fair comparison, we measure the amortized latency–the time required for link predictions of a single node. As reported in Table 3, our method achieves an amortized latency of 0.24s on Cora, which is $5.92\times$, $4.83\times$, $3.25\times$ and $2.67\times$ faster than that of Gazelle, E2DM, HElayers and uSCORE, respectively. The similar improvement can be observed across the Citeseer and PubMed.

The superiority of our method over other SOTA solutions can be attributed to two key factors:

The first reason is that all prior approaches are primarily designed to solve the single matrix-matrix multiplication problem instead of two-way matrix multiplications in GCN inference. In GCN inference, the computation pattern mainly consists of two computational components-the common FC layer for graph feature aggregation ($X \cdot W$) and the adjacency matrix multiplication for graph node aggregation ($A \cdot X$). These two components represent distinct and orthogonal computation patterns for the input feature map. Consequently, optimizing only one of these directions leads to sub-optimal performance. In contrast, our proposed "Penguin", which incorporates two-way parallel encoding and an optimized sub-block matrix size, offers superior performance with theoretical guarantees.

The second compelling factor contributing to the excellence of our approach is the adaptability to address the issue of wasted slots of ciphertext, a capability notably absent in prior methods. In studies such as [15, 1, 13], traditional matrix-multiplication methods establish sub-block matrix partitions based on square or rectangular sub-block matrices determined by the input weight matrix size. However, these conventional methods face challenges in effectively managing the problem of wasted slots that arise after feature aggregation in Graph Convolutional Network (GCN) inference. As confirmed in Table 3, our two-way parallel packing method substantially improves HE-GCN inference by leveraging the Interleaved Assembling technique that can effectively minimize the wasted slots. This, in turn, leads to extra memory space and HOC reduction.

Putting all these together, our method clearly demonstrates its advantage over SOTA solutions in accelerating HE-GCN inference.

# 5  Related Work

CryptoNets [10] is the first work that demonstrates the feasibility of building privacy-preserving machine learning (PPML) by HE. However, the long inference latency and the inflexible packing format make it hard to be applied to large-scale models and datasets. Another following work named SHE [25], translates the nonlinear ReLU and Max Pooling operations as Boolean operations to support the TFHE-based [7] PPML without modifying the pretrained models. There also exist many multi-party computation (MPC) solutions that combine the two-party computation protocols [36] with HE frameworks to achieve the low inference latency [30, 17, 26, 24, 27]. However, they suffer from high communication overhead incurred by data transfer between multiple parties. Recent studies such as LoLa [4], CHET [8], and HEAR [20] leverage the ciphertext packing technique to place multiple data in the same ciphertext so that HE operations can be conducted efficiently via single instruction multiple data (SIMD) for accelerating HE-based CNN inference. These approaches are often not applicable or optimal to GCN inference due to the very different computation patterns between the GCN and CNN. CryptoGCN [28] is the first attempt to build HE-based PPML for GCNs. It packs the ciphertexts from individual node to relieve the adjacency matrix multiplication overhead. However, they assume the adjacency matrix as plaintext, which is not applicable to dynamic graph settings which require protecting both graph structure and features like our work.

Cheetah [14] aims to eliminate the need of costly rotation in HE-matrix multiplication. The basic idea is to replace the dot-sum operation with two polynomials multiplications. However, it requires communication between the client and server to perform ciphertext decryption and re-encryption after each layer computation, e.g. after a convolutional layer. In contrast, our approach operates within a "HE without-client-aid" setting. Here, the server only requires the client to encrypt and send data to the server once, after which the server can perform the whole computation of inference and only send the final encrypted result back to the client. It does not require the frequent communication between client and server like [14]. Therefore, our work focuses on HE computation optimization. In our view, our solution and [14]'s MPC+HE setting represent two distinct directions to realize PPML. Each approach is tailored for different private inference scenarios. Hence, conducting a direct comparison between our method and the approach in [14] could be challenging since they are designed for different purposes and provide distinct advantages.

Gazelle [17] is a hybrid matrix multiplication algorithm based on mixture of naive implementation and diagonal encoding method [12]. It is designed to handle the general matrix multiplication that occurs in FC layers, where the output dimension is smaller than the input dimension. Additionally, E2DM [15] and uSCORE [13] address encrypted matrix-matrix multiplication optimization by breaking down the problem into small square matrix multiplications and facilitating consecutive matrix multiplications.HElayers [1], on the other hand, leverages a tiles-tensor-based matrix-multiplication technique for multiplying two square matrices and consumes less multiplicative depth compared to [15]. While these general solutions have proven effective in various contexts, their efficiency is limited when applied to accelerate Homomorphic Encryption-based Graph Convolutional Network (HE-GCN) inference, as detailed in Section 4.2.

# 6  Conclusion

In this paper, we propose a two-dimension parallel packing technique for feature ciphertext by optimizing the feature matrix partition size and further propose an interleaved assembling technique to merge ciphertexts that have wasted slots from feature reduction in CKKS-based secure GCN inference. These techniques can better save ciphertext memory and effectively reduce the number of homomorphic operations required. Experimental results based on the GAEs for link prediction and 3 popular graph datasets show that our solution can speed up the latency of the secure GCN inference by $10\times$ and reduce the memory requirement by more than 79%, greatly outperforming the state-of-the-art solutions.

# 7  Acknowledgment

We thank all anonymous reviewers for their constructive comments and suggestions on this work. This work is partially supported by the National Science Foundation (NSF) under Grants No. CNS-2348733 and CNS- 2349538.

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
