we discussed for the large graph (see Section 3.2), all designs perform much worse in PubMed than the other two datasets. This is because the number of nodes in PubMed is significantly larger than the size of ciphertext ($19717 \gg 4096$), which means that it needs to be multiplied with a large $19717 \times 19717$ adjacency matrix. Therefore, the number of CMult $\gg$ number of Rot. However, our proposed packing technique can still improve the performance in such cases.

#### 4.2.2 Interleaved Assembling

Table 2 also reports the evaluation results of incorporating the two-dimensional parallel packing and interleaved assembly methods. For example, in Cora, the number of rotations, the number of CMult, and the number of other HOCs in the $Penguin(32, 128) + IA$ design are further reduced by 1.4K, 114.7K, and 31K, respectively, compared to the parallel-packing only $Penguin(32, 128)$. This makes $Penguin(32, 128) + IA$ the best design on all datasets, i.e., with the minimum memory usage of 0.49GB, 1.39GB, and 3.76GB and a $10.62\times$, $9.96\times$, and $1.46\times$ speedup on dataset Cora, Citeseer, and PubMed, respectively. These results illustrate that our proposed interleaved assembly can effectively reduce the wasted empty slots and save the number of ciphertexts selected in the computation, thus significantly improving the efficiency based on the SIMD.

Table 3: Compare with the state-of-the-art.

| Dataset | Method | Security Level | Latency (s) | Amortized Latency | Speedup ($\times$) |
|---------|--------|----------------|-------------|-------------------|-------------------|
| Cora | E2DM(64) [14] | 98-bit | 3150.74 | 1.16 | - |
| | uSCORE(32,128) [12] | 98-bit | 1727.12 | 0.64 | 1.82 |
| | **Penguin(32,128)+IA** | 128-bit | **660.57** | **0.24** | **4.77** |
| Citeseer | E2DM(64) [14] | 98-bit | 4561.15 | 1.37 | - |
| | uSCORE(32,128) [12] | 98-bit | 2377.50 | 0.72 | 1.92 |
| | **Penguin(32,128)+IA** | 128-bit | **928.10** | **0.28** | **4.91** |
| Pubmed | E2DM(64) [14] | 98-bit | 154530.49 | 7.84 | - |
| | uSCORE(32,128) [12] | 98-bit | 78843.49 | 4.00 | 1.96 |
| | **Penguin(32,128)+IA** | 128-bit | **30522.43** | **1.55** | **5.06** |

### 4.2.3 Compare with SOTA Solutions

In our evaluation, we also compare our best design $Penguin(32, 128) + IA$ with the state-of-the-art (SOTA) solutions, including E2DM [14] and uSCORE [12]. Both SOTA solutions can speed up HE-GCN inference using the optimized matrix-matrix multiplication. Table 3 reports the results. Our encryption parameters can guarantee a 128-bit security level, which is higher than SOTA solutions that need more multiplicative levels to mask the plaintexts. To provide a fair comparison, we measure amortized latency, which is the latency required for link predictions of one node. As listed in Table 3, our method achieves an amortized latency of 0.24s on the Cora, which is $4.77\times$ (or $1.82\times$) faster than that of E2DM (or uSCORE). We observe a similar improvement on the Citeseer and PubMed. These results illustrate that by leveraging the unique features of GCN computation to reduce the number of ciphertexts and HOCs, our method significantly outperforms the SOTA methods that are based on the optimization of the general matrix-matrix multiplication in encryption domain.

## 5 Related Work

CryptoNets [9] is the first work that demonstrates the feasibility of building privacy-preserving machine learning (PPML) by HE. However, the long inference latency and the inflexible packing format make it hard to be applied to large-scale models and datasets. Another following work named SHE [24], translates the nonlinear ReLU and Max Pooling operations as Boolean operations to support the TFHE-based [6] PPML without modifying the pretrained models. There also exist many multi-party computation (MPC) solutions that combine the two-party computation protocols [35] with HE frameworks to achieve the low inference latency [29, 16, 25, 13, 23, 26]. However, they suffer from high communication overhead incurred by data transfer between multiple parties. Recent studies such as LoLa [3], CHET [7], and HEAR [19] leverage the ciphertext packing technique to place multiple data in the same ciphertext so that HE operations can be conducted efficiently via single instruction multiple data (SIMD) for accelerating HE-based CNN inference. These approaches are often not applicable or optimal to GCN inference due to the very different computation patterns between the GCN and CNN. CryptoGCN [27] is the first attempt to build HE-based PPML for GCNs. It packs the ciphertexts from individual node to relieve the adjacency matrix multiplication overhead. However, they assume the adjacency matrix as plaintext, which is not applicable to dynamic graph settings which require protecting both graph structure and features like our work. E2DM [14] and uSCORE [12] consider the two encrypted matrix-matrix multiplication optimization by decomposing the problem into small square matrix multiplication and supporting the consecutive matrix multiplication. However, these general solutions demonstrate limited efficiency to accelerate HE-based GCN inference, as shown in Sec. 4.2.

## 6 Conclusion

In this paper, we propose a two-dimension parallel packing technique for feature ciphertext by optimizing the feature matrix partition size and further propose an interleaved assembling technique to merge ciphertexts that have wasted slots from feature reduction in CKKS-based secure GCN inference. These techniques can better save ciphertext memory and effectively reduce the number of homomorphic operations required. Experimental results based on the GAEs for link prediction and 3 popular graph datasets show that our solution can speed up the latency of the secure GCN inference by $10\times$ and reduce the memory requirement by more than 79%, greatly outperforming the state-of-the-art solutions.