# OpenReview forum: "Penguin: Parallel-Packed Homomorphic Encryption for Fast Graph Convolutional Network Inference"
_NeurIPS.cc/2023/Conference — NeurIPS 2023 poster_

### Official Review · Reviewer_idN7 · 2023-07-05

**Soundness:** 3 good
**Presentation:** 3 good
**Contribution:** 3 good
**Rating:** 6
**Confidence:** 4

**Summary:**

To alleviate the dramatic computation and memory overhead in HE-based GCN inference, this paper proposes a new HE-based ciphertext packing technique named Penguin. Penguin focuses on a sequence of matrix-matrix multiplications which is the bottleneck during private GCN inference. Thus, it employs an effective two-dimension parallel packing technique and an interleaved assembly technique to reduce the number of HE rotations while make use of the blank slots in polynomials. This paper also provides theoretical analysis and experimental validation to demonstrate the speedup achieved by Penguin in accelerating private GCN inference. Results show that Penguin can achieve up to ∼ 10× speedup and around ∼ 79% reduction in computational memory overhead, outperforming SOTA solutions.

**Strengths:**

1. Reasonable motivation. This paper first analyzes the latency breakdown in private GCN inference and finds that the bottleneck is a sequence of matrix-matrix multiplications $A\cdot X\cdot W$ due to a great deal of Rotations and CMults. Additionally, the inefficient packing method is the deeper reason. Therefore, this paper proposes several packing techniques to solve this issue.

2. Direct and effective method. Based on the node-wise and feature-wise packing, this paper proposes Two-Dimension Parallel-Packing which selects tiling size to minimize the number of rotations. What's more, when feature size is reduced in aggregation, Interleaved Assembling can be used to make a denser packing. Experiments demonstrate the effectiveness of the two methods.

3. Well written. The writing quality of the paper is excellent, with logical coherence.


**Weaknesses:**

1. Missing comparison with several highly related works. The node-wise and feature-wise packing methods described in Sec 3 are actually common in SIMD-based HE methods. Thus, I think this paper needs to have a comparison with SIMD-based work like Gazelle [1].

[1] Juvekar, Chiraag, Vinod Vaikuntanathan, and Anantha Chandrakasan. "{GAZELLE}: A low latency framework for secure neural network inference." 27th USENIX Security Symposium (USENIX Security 18). 2018.

[2] Reagen, Brandon, et al. "Cheetah: Optimizing and accelerating homomorphic encryption for private inference." 2021 IEEE International Symposium on High-Performance Computer Architecture (HPCA). IEEE, 2021.

[3] Hao, Meng, et al. "Iron: Private inference on transformers." Advances in Neural Information Processing Systems 35 (2022): 15718-15731.

2. There exists another type of method named coefficient encoding which has extremely high performance in matrix-matrix and matrix-vector multiplication[2,3]. So the SOTA methods claimed by the author are questionable.

3. The baseline of the paper is not very strong. How does the proposed method compare to CryptoGCN and other works that directly optimize HE for GCN instead of CNN?

**Questions:**

1. How does the proposed method compare to the coefficient encoding-based method like Cheetah?

2. How about the comparison with stronger baselines like CryptoGCN that optimizes HE for GCN?

3. In Table 2, Penguin(32,128)+IA significantly reduces the number of CMults over 10 times, but the latency only reduces 20% compared to Penguin(32,128). Why is this situation happening?

---

> ### Author Rebuttal · Authors · 2023-08-10
>
> **1. Response to Weakness 1-Missing Comparison:**
>
> Thanks for your constructive feedback. We have conducted additional experiments and reported the comparison results with Gazelle [1] in Table 1 below. As Table 1 shows, our proposed solution outperforms that of Gazelle across all three datasets. The reason is as follows:  the proposed hybrid approach in Gazelle[1] is mainly designed to solve the single matrix-multiplication problem in the general FC layer (input feature > output feature). However, GCN inference contains two parts of computation-one is FC layer (XW) and the other is adjacency matrix multiplication (AX) that is a square matrix (fixed size). When computing with an adjacency matrix, the proposed method in [1] would not optimize its packing but instead just adopt the diagonal wise-encoding method from [2]. Thus, our proposed Penguin-two-way parallel-encoding with optimized sub-block matrix size could perform better here.
>
> **Table 1**
> |  | Cora | Citeseer | PubMed |
> |---|---|---|---|
> | Gazelle| 3832.36s| 4727.94s| 158655.54s|
> | Penguin(32,128)+IA| 660.67s| 928.1s| 30522.43s|
>
> **2. Response to Weakness 2 and Question 1-Why CKKS not coefficient encoding method [3]:**
>
> Thanks for your constructive comments regarding coefficient-encoding work. There are several differences between our solution and the referred coefficient encoding work. First,
> our work has a different threat model setting compared to [3]. The coefficient encoding rotation-free HE in [3] requires clients assistance for private inference, where our work assumes the client doesn’t have much computation capability and would not participate in the inference computation process. The method used in [3] is BFV, in which it requires the client to do the decryption for the extracted LWE ciphertext, re-encryption for intermediate results and then send the new ciphertexts to the server, then the server can perform the next layer’s computation. As a result, communication is the bottleneck in their proposed method [3]. However, in our CKKS HE without-client-aid setting, the server side only requires the client to encrypt and send data to the server once, and the server can perform the computation one time and only send the final encrypted result to the client. It does not require the frequent communication between client and server like [3]. Therefore, HE computation, instead of communication, becomes the bottleneck. In our view, our HE without-client-aid setting and [3]’s MPC+HE setting represent two orthogonal directions to realize PPML, and they are suitable for different private inference scenarios. Our paper aims at reducing the server’s computation overhead under the HE without-client-aid setting. It is very difficult to directly compare these two approaches, as both have pros and cons under different assumptions. In the future, it would be interesting to explore the coefficient encoding method for HE-based GCN inference. We will include such discussion in the related work based on the reviewer’s further suggestions.
>
> **3. Response to Weakness 3 and Question 2-Comparison with CryptoGCN:**
>
> Thanks for your comments. Our experiments have included a fair comparison with CryptoGCN. For CryptoGCN, the key idea is to use Adjacency Matrix Aware (AMA) ciphertext encoding technique, followed by the patterned sparse matrix partitioning to take advantage of the sparsity of unencrypted adjacency matrix A, so as to reduce the costly HE operations in GCN.
> While deploying AMA encoding technique is still applicable when both graph node features and structure are encrypted, we would like to emphasize that the patterned sparse matrix partitioning technique is not applicable since here the adjacency matrix A is encrypted and the sparsity cannot be exploited. Therefore, to make our work comparable with CryptoGCN under the same setting (both encrypted features and adjacency matrix), in our experiments, we implemented CrytoGCN’s AMA encoding format as the node-wise format which encodes the features from the same node into one ciphertext (See Table 2 in our paper,  Cora-Penguin(1433,1), Citeseer-Penguin(3703,1), and PubMed-Penguin(19717,1)). As Table 2 shows, the performance of our method (Penguin 32,128) is much better than those with the AMA encoding from CryptoGCN. The reason is twofold: 1) encrypted adjacency matrix does not offer the public sparsity information that can be exploited by CryptoGCN; 2) two-dimensional optimization (feature-node AXW) instead of CryptoGCN’s AMA encoding focusing on one-direction optimization (adjacency matrix multiplication AX) only.
>
>
> **4. Response to Question 3-concern about the weird data**
>
> Thanks for your careful review. It is 93K instead of 9.3K. Sorry about the typo. We will fix this part in future revision of our manuscript.
>
> Reference:
>
> [1]  Juvekar, Chiraag, Vinod Vaikuntanathan, and Anantha Chandrakasan. "{GAZELLE}: A low latency framework for secure neural network inference." 27th USENIX Security Symposium (USENIX Security 18). 2018.
>
> [2] Halevi, Shai, and Victor Shoup. "Algorithms in helib." Advances in Cryptology–CRYPTO 2014: 34th Annual Cryptology Conference, Santa Barbara, CA, USA, August 17-21, 2014, Proceedings, Part I 34. Springer Berlin Heidelberg, 2014.
>
> [3] Huang, Zhicong, et al. "Cheetah: Lean and fast secure {two-party} deep neural network inference." 31st USENIX Security Symposium (USENIX Security 22). 2022.

---

### Official Review · Reviewer_MKSk · 2023-07-05

**Soundness:** 2 fair
**Presentation:** 2 fair
**Contribution:** 2 fair
**Rating:** 5
**Confidence:** 2

**Summary:**

The paper introduces Penguin, a novel HE-based ciphertext packing technique for accelerating GCN inference on encrypted graph data while ensuring data privacy. By exploiting the unique computation pattern of GCN layers, Penguin reduces computation and memory overhead associated with HE operations. The technique achieves significant speedup and reduction in computational memory overhead compared to state-of-the-art solutions. This work is the first to address the protection of both graph structure and features in accelerating HE-GCN inference on encrypted data.

**Strengths:**

Introduces Penguin, a novel HE-based ciphertext packing technique for accelerating GCN inference on encrypted graph data.
Exploits the computation pattern of GCN layers to reduce computation and memory overhead associated with HE operations.
Provides theoretical analysis and experimental validation to demonstrate the speedup achieved by Penguin.
Achieves significant speedup and reduction in computational memory overhead compared to state-of-the-art solutions.
Addresses the protection of both graph structure and features in HE-GCN inference on encrypted data.

**Weaknesses:**

It would be beneficial to provide more detailed comparisons with existing approaches to highlight the advantages of Penguin.

**Questions:**

-

**Limitations:**

The paper could benefit from further discussion on the limitations and potential future directions of the proposed technique.

---

> ### Author Rebuttal · Authors · 2023-08-10
>
> **Response to Weakness- More comparisons with other existing works:**
>
> Thanks for your comments. In addition to the baselines E2DM[3], uSCORE[4], We have conducted additional experiments and reported the results comparisons with other existing relevant approaches (e.g. Gazelle [1] and HElayers [2]) in the following Table. In particular, HElayers [2] , a state-of-the-art HE Packing method suggested by Reviewer pXg9, Gazelle [1] is another work aiming to address the single matrix-multiplication problem in the general FC layer, suggested by Reviewer idN7. As Table 1 shows, our Penguin still achieves the best latency performance because it takes advantage of the unique computation pattern during the HE-GCN inference to significantly reduce the computation and memory overhead associated with HE operations. Following is the table showing the comparison results:
>
> **Table 1**
> |  | Cora | Citeseer | PubMed |
> |---|---|---|---|
> | Gazelle| 3832.36s| 4727.94s| 158655.54s|
> | HElayers | 2102.47s| 3044.58s| 103283.56s|
> | Penguin(32,128)+IA| 660.67s| 928.1s| 30522.43s|
>
>
>
> References:
>
> [1]Juvekar, Chiraag,Vinod Vaikuntanathan, and Anantha Chandrakasan. "{GAZELLE}: A low latency framework for secure neural network inference." 27th USENIX Security Symposium (USENIX Security 18). 2018.
>
> [2]Aharoni et al., "HElayers: A tile tensors framework for large neural networks on encrypted data, " PoPETs 2023.
>
> [3] Jiang, Xiaoqian, et al. "Secure outsourced matrix computation and application to neural networks." Proceedings of the 2018 ACM SIGSAC conference on computer and communications security. 2018.
>
> [4] Huang, Zhicong, et al. "More efficient secure matrix multiplication for unbalanced recommender systems." IEEE Transactions on Dependable and Secure Computing (2021).

---

### Official Review · Reviewer_pXg9 · 2023-07-08

**Soundness:** 3 good
**Presentation:** 4 excellent
**Contribution:** 2 fair
**Rating:** 7
**Confidence:** 3

**Summary:**

This paper proposed an efficient data-packing method for cryptographically-secure inference on GCN, where the feature matrix and adjacency matrix are encrypted using homomorphic encryption (HE). The problem statement is interesting as the GCNs typically exhibit a significant sparsity level, increasing the number of rotations and, consequently, causing an increase in the HE latency.





**Strengths:**

1. The proposed solution is *novel* for dealing with rotation inefficiency in HE, which stems from the high degree of sparsity in GCN.


2. The presentation of the proposed solution is excellent, and the paper provides clear and detailed information about the experimental methodology, including the specific parameters of HE.

**Weaknesses:**


**Comparison with SOTA data packing method**
Table 3 in the paper does not compare the proposed solution and the state-of-the-art HE-packing method [1]. It is worth noting that the HELayer [1] incorporates a packing optimizer that selects the most efficient packing method to optimize latency and memory usage in HE convolution operations. Conducting a study on the suitability of the HELayer packing for graph convolution and their comparison with the proposed solution can provide valuable insights.

**Lack of experiments for showing the impact of sparsity on the efficacy of the proposed packing method**
An experimental study and discussion on the impact of sparsity on the effectiveness of existing/state-of-the-art HE packing methods versus the proposed methods in GCNs would provide valuable insights.


**CryptoGCN + Penguin**
Including an experimental analysis in the paper would have been beneficial to understand how the proposed approach in CryptoGCN (NeurIPS'22) for reducing the multiplicative depth can further enhance the efficiency of the proposed HE packing method.


1. Aharoni et al., "HElayers: A tile tensors framework for large neural networks on encrypted data, " PoPETs 2023.


**Questions:**

Can we employ the rotation-free HE implementation [1] for faster private inference on GCNs? How does the sparsity in GCN impact the solution proposed for making HE rotation free [1]?




1.  Huang et al., "Cheetah: Lean and fast secure two-party deep neural network inference," USENIX Security 2022.

**Limitations:**

Not addressed in the paper.

---

> ### Author Rebuttal · Authors · 2023-08-10
>
> **1. Response to Weakness 1-Comparison with SOTA data packing method:**
>
> Thanks for your constructive comments. We have conducted the experiments and reported the comparison results with HElayers in Table 1 below.  As Table 1 shows, our solution–Penguin beats HElayers consistently across the datasets because:  1) Principally, HElayers [2] focuses on the computation of a fully-connected layer which is a one-direction (single) matrix multiplication. However, GCN inference has different bottlenecked computation patterns, e.g. two-way matrix multiplication (AXW) in each layer. Our proposed method targets an optimization problem for a two-way matrix-matrix multiplication (adjacency matrix multiplication and fully-connected layer weight matrix multiplication) and comes up with the best matrix blocking size with theoretical guarantee. 2) In HElayers, their matrix-multiplication method has the same computation complexity as E2DM [3] (in section 8.1 of [2]) and also requires a series of fixed square sub-blocks matrices to replace the original matrix multiplication. Thus,  similar to E2DM (a benchmark we compared in Table 3 in our paper), the proposed matrix-multiplication method in [2] cannot handle the problem of the wasted slots. Therefore, as expected in Table 1, our two-way parallel packing can achieve much better HE-GCN inference than that of HElayers. We will incorporate such discussions in the experiments based on the reviewer’s advice.
>
> **Table 1**
> |  | Cora | Citeseer | PubMed |
> |---|---|---|---|
> | HElayers | 2102.47s| 3044.58s| 103283.56s|
> | Penguin(32,128)+IA| 660.67s| 928.1s| 30522.43s|
>
>
> **2. Response to Weakness 2-Lack of experiments for showing the impact of sparsity on the efficacy of the proposed packing method**
>
> We appreciate your question regarding the impact of sparsity on the efficacy of the packing method. However,  we would like to clarify that in this work, we assume the adjacency matrices (A) are encrypted and the actual element value in the matrix could not be seen by the server. In other words, both adjacency matrix A and feature matrix X are encrypted, which is different from that of CryptoGCN-adjacency matrix A is a plaintext matrix whose sparsity can be exploited along with ciphertext packing for speedup. Thus, we can not leverage the sparsity of the adjacency matrix to skip redundant HE operations like CryptoGCN does.
>
> **3. Response to Weakness 3-CryptoGCN + Penguin**
>
> Thanks for your suggestion. Actually we have tried to reduce the activation layers of our evaluated GNN models for multiplicative depth reduction and the corresponding accuracy and latency results are reported in Table 2 and 3 below respectively. As Table 2 shows, the accuracy drops when pruning 2 activations, especially for the Citeseer dataset. If we prune two activation layers, the saved two levels would reduce the requirement of encryption parameter ciphertext modulus Q from 218 to 158. However, this could not allow us to change the polynomial degree from 8192 to 4096 like CryptoGCN  because we still have to use N=8192 to maintain at least 128 bits security level for current Q [1]. As Table 3 illustrates, it can still achieve latency reduction but the improvement is not as significant as that of CryptoGCN despite the more prominent accuracy drop.
>
> **Table 2**
> |  | Cora | Citeseer | PubMed |
> |---|---|---|---|
> | No activation pruning | 0.974 | 0.747 | 0.858 |
> | With 2 activations pruned| 0.958 | 0.659 | 0.855 |
>
>
> **Table 3**
> |  | Cora | Citeseer | PubMed |
> |---|---|---|---|
> | Q=218 | 660.67s| 928.1s| 30522.43s|
> | Q=188 | 453.56s | 642.4s| 20855.38s|
> | Q=158 | 353.07s| 501.51s| 16215.66s|
>
> **4. Response to Question 1-Rotation-free approaches**
>
> Thanks for your interesting question. The rotation-free implementation by [1] is in a MPC+HE setting. It needs the client to do decryption and re-encryption for completing the computation process to achieve the rotation-free. In this way, the computation latency on the server can be low. However, this method comes at the cost of communication between server and client (one-layer computation with one-time communication). Also, the client needs to frequently decrypt and then re-encrypt the intermediate results sent by the server. In our work, we assume a HE without-client-aid setting and do not require the client to interact with the server during the inference. The setting and application scenario are different from that of MPC + HE. In our setting, the HE computation at the server, instead of communication, becomes the bottleneck. Therefore, we think the two approaches represent different directions to achieve the privacy-preserving machine learning applicable to different scenarios, and both solutions have pros and cons. In our view, it would be interesting to explore such a rotation-free solution for private GCN inference in future work.
>
> In regard to the sparsity’s impact on rotation-free solution, our understanding is that if the assumption for the adjacency matrix is in non-encrypted status, we think it is still possible to use sparsity skip some operations, like two polynomial multiplications. However, in our work here, we would like to clarify that we cannot leverage the sparsity to skip HE operations as the adjacency matrix is encrypted.
>
> Reference:
>
> [1] Laine, Kim. "Simple encrypted arithmetic library 2.3. 1." Microsoft Research https://www. microsoft. com/en-us/research/uploads/prod/2017/11/sealmanual-2-3-1. pdf (2017).
>
> [2] Aharoni et al., "HElayers: A tile tensors framework for large neural networks on encrypted data, " PoPETs 2023.
>
> [3] Jiang, Xiaoqian, et al. "Secure outsourced matrix computation and application to neural networks." Proceedings of the 2018 ACM SIGSAC conference on computer and communications security. 2018.

---

> > ### Comment · Reviewer_pXg9 · 2023-08-11
> > **Response to rebuttal**
> >
> >
> > Thank you for the detailed rebuttal. Given a tight timeline, I appreciate Auhtor's effort in presenting additional experimental data, especially a comparison with HELayers, and discussing the shortcomings of the rotation-free HE implementation. I would encourage the authors to include these data and discussions in the updated version of the paper. Based on the rebuttal, I increased the score to Accept.

---

> > > ### Author Response · Authors · 2023-08-11
> > > **Response to Reviewer pXg9**
> > >
> > > We sincerely thank for your very helpful comments and strong support on our work. We will incorporate the discussion in the paper as you suggested. Thanks again!

---

### Official Review · Reviewer_RE11 · 2023-07-10

**Soundness:** 3 good
**Presentation:** 3 good
**Contribution:** 3 good
**Rating:** 6
**Confidence:** 3

**Summary:**

The paper proposes a framework for optimizing latency in secure inference of GNNs. The authors propose a CKKS based packing scheme that is tailored for the structure of operations applied in Graph Convolution Networks. This appears to be the first work that considers both adjacency matrix and the feature matrix as private.

**Strengths:**

1. The optimizations for parallel packing introduced in this work are well motivated and justified with experimental data.
2. The technique provides a significant latency improvement over existing matrix multiplication techniques empirically.
3. I found the overall in-depth analysis of GCN computation and utilizing it for optimizing packing and reducing wasted operations with interleaved assembling to be well designed.

**Weaknesses:**

1. I wonder if a fair evaluation would include cryptoGCNs with encrypted adjacency matrix.
2. The latency is extremely high for practical purposes, however, there isn't much work in this specific problem therefore it is difficult to understand the difficulty of the task.
3. How does this technique compare with SecGNN[1]? (Ignoring the training aspect)

References -
1. Wang, Songlei, Yifeng Zheng, and Xiaohua Jia. "SecGNN: Privacy-preserving graph neural network training and inference as a cloud service." IEEE Transactions on Services Computing (2023).

**Questions:**

None

**Limitations:**

I could not find any discussion on limitations by the authors even though they claim it in the submission form.

---

> ### Author Rebuttal · Authors · 2023-08-10
>
> **1. Response to Weakness 1-if a fair evaluation would include CryptoGCN:**
>
> Thanks for your comments. Our experiments have included a fair comparison with CryptoGCN. For CryptoGCN, the key idea is to use Adjacency Matrix Aware (AMA) ciphertext encoding technique, followed by the patterned sparse matrix partitioning to take advantage of the sparsity of unencrypted adjacency matrix A, so as to reduce the costly HE operations in GCN.
> While deploying AMA encoding technique is still applicable when both graph node features and structure are encrypted, we would like to emphasize that the patterned sparse matrix partitioning technique is not applicable since here the adjacency matrix A is encrypted and the sparsity cannot be exploited. Therefore, to make our work comparable with CryptoGCN under the same setting (both encrypted features and adjacency matrix), in our experiments, we implemented CrytoGCN’s AMA encoding format as the node-wise format which encodes the features from the same node into one ciphertext (See Table 2,  Cora-Penguin(1433,1), Citeseer-Penguin(3703,1), and PubMed-Penguin(19717,1)). As Table 2 shows, the performance of our method (Penguin 32,128) is much better than those with the AMA encoding from CryptoGCN. The reason is twofold: 1) encrypted adjacency matrix does not offer the public sparsity information that can be exploited by CryptoGCN; 2) two-dimensional optimization (feature-node AXW) instead of CryptoGCN’s AMA encoding focusing on one-direction optimization (adjacency matrix multiplication AX) only.
>
> **2. Response to Weakness 2-concern about the difficulty about our research problem:**
>
> Homomorphic Encryption (HE) is a promising technology to realize the PPML inference. Also the well-known challenge of HE is the large computation and memory (thus latency) overhead especially facing machine learning.This leads to the prolonged inference latency which greatly hinders its practicality at the current stage, as illustrated in many prior HE-based PPML for Deep CNN models, e.g. ResNet-20 [4,5]. Compared with HE-based CNNs, HE-based GCN is an emerging area that has been far less explored so far. To the best of our knowledge, the SOTA work for HE-based GCN inference-CryptoGCN requires an inference latency 4273.89s for 25✖25 adjacency matrix due to the unique and expensive matrix-matrix multiplications in the encryption domain. Compared with CryptoGCN, our work involves much larger adjacency matrices (e.g. 19717✖19717), of which the computation and latency overhead is expected to be much higher. While our work has not yet reached the latency requirement of practical applications, it represents an early attempt along this challenging direction. We believe there is room for further performance improvement from the aspects of algorithm and hardware. For example, with the advancement of dedicated HE hardware accelerators [2,3], the HE operations latencies could be further reduced.
>
> **3. Response to Weakness 3-comparison with SecGNN:**
>
> SecGNN is based on a technique called additive secret-sharing in a three-servers setting where the overhead mainly stems from the communication between different servers, as it involves the interaction among servers. However, for our HE without-client-aid setting (no communication between server and client during inference), the overhead is mainly from the HE operations (e.g. rotation and ciphertext-ciphertext multiplication) for completing the required computations on the server. These two methods represent two directions for achieving PPML under different application scenarios or settings, of which the former is bottlenecked by communication while the latter is bottlenecked by computation. Therefore, we believe it is difficult to directly compare them.
>
> Reference:
>
> [1] Lee, Eunsang, et al. "Low-complexity deep convolutional neural networks on fully homomorphic encryption using multiplexed parallel convolutions." International Conference on Machine Learning. PMLR, 2022.
>
> [2] Samardzic, Nikola, et al. "F1: A fast and programmable accelerator for fully homomorphic encryption." MICRO-54: 54th Annual IEEE/ACM International Symposium on Microarchitecture. 2021.
>
> [3] Kim, Sangpyo, et al. "Bts: An accelerator for bootstrappable fully homomorphic encryption." Proceedings of the 49th Annual International Symposium on Computer Architecture. 2022.
>
> [4] Lee, Eunsang, et al. "Low-complexity deep convolutional neural networks on fully homomorphic encryption using multiplexed parallel convolutions." International Conference on Machine Learning. PMLR, 2022.
>
> [5] Ran, Ran, et al. "SpENCNN: Orchestrating Encoding and Sparsity for Fast Homomorphically Encrypted Neural Network Inference." International Conference on Machine Learning. PMLR, 2023.

---

### Decision · Program_Chairs · 2023-09-21

**Decision:**

Accept (poster)

**Comment:**

The paper proposes an efficient HE-based GCN inference method, by exploiting the unique computation patterns of GCN layers.

Reviewers recognize the importance of the problem as well as the technical contributions. The authors addressed most of the concerns raised by the reviewers. They have also added additional experiments of comparison with the SOTA data packing method. The reviewers unanimously recommended accept after the rebuttal (6, 6, 5, 7).

Please add the clarifications and additional experiments in the final version.